

# Comparative functional genomic analysis of Alzheimer's affected and naturally aging brains

Yi-Shian Peng[1], Chia-Wei Tang[1], Yi-Yun Peng[1], Hung Chang[1], Chien-Lung Chen[2], Shu-Lin Guo[3,4], Li-Ching Wu[1], Min-Chang Huang[5] and Hoong-Chien Lee[1,5]

[1] Department of Biomedical Sciences and Engineering, National Central University, Taoyuan, Taiwan
[2] Department of Nephrology, Landseed Hospital, Taoyuan, Taiwan
[3] Department of Anesthesiology, Cathay General Hospital, Taipei, Taiwan
[4] Department of Anesthesiology, Tri-Service General Hospital and National Defense Medical Center, Taipei, Taiwan
[5] Department of Physics, Chung Yuan Christian University, Taoyuan, Taiwan

## ABSTRACT

**Background.** Alzheimer's disease (AD) is a prevalent progressive neurodegenerative human disease whose cause remains unclear. Numerous initially highly hopeful anti-AD drugs based on the amyloid-β (Aβ) hypothesis of AD have failed recent late-phase tests. Natural aging (AG) is a high-risk factor for AD. Here, we aim to gain insights in AD that may lead to its novel therapeutic treatment through conducting meta-analyses of gene expression microarray data from AG and AD-affected brain.

**Methods.** Five sets of gene expression microarray data from different regions of AD (hereafter, ALZ when referring to data)-affected brain, and one set from AG, were analyzed by means of the application of the methods of differentially expressed genes and differentially co-expressed gene pairs for the identification of putatively disrupted biological pathways and associated abnormal molecular contents.

**Results.** Brain-region specificity among ALZ cases and AG-ALZ differences in gene expression and in KEGG pathway disruption were identified. Strong heterogeneity in AD signatures among the five brain regions was observed: HC/PC/SFG showed clear and pronounced AD signatures, MTG moderately so, and EC showed essentially none. There were stark differences between ALZ and AG. OXPHOS and Proteasome were the most disrupted pathways in HC/PC/SFG, while AG showed no OXPHOS disruption and relatively weak Proteasome disruption in AG. Metabolic related pathways including TCA cycle and Pyruvate metabolism were disrupted in ALZ but not in AG. Three pathogenic infection related pathways were disrupted in ALZ. Many cancer and signaling related pathways were shown to be disrupted AG but far less so in ALZ, and not at all in HC. We identified 54 "ALZ-only" differentially expressed genes, all down-regulated and which, when used to augment the gene list of the KEGG AD pathway, made it significantly more AD-specific.

Corresponding authors
Yi-Shian Peng,
bim962511@gmail.com
Hoong-Chien Lee,
hclee12345@gmail.com

## INTRODUCTION

Aging (AG), or senescence, is a natural process that leads to deterioration in biological, physical, mental, and neurodegenerative disorders including dementia (*López-Otín et al., 2013*). Alzheimer's disease (AD), a neurodegenerative disorder with millions of individuals affected worldwide, is the most common type of dementia (*Wang et al., 2009a*).

The etiology of AD is not well understood, but about 70% of the risk for AD is thought to be involving many genes. In the last 25 years, the amyloid cascade, or amyloid-β (Aβ), hypothesis (*Hardy & Higgins, 1992*; *Hardy & Selkoe, 2002*), which holds that Aβ aggregation in the brain is a main causative factor of AD and mutations in presenilin 1 (*PSEN1*) (*Kelleher & Shen, 2017*), presenilin 2 (*PSEN2*) (*Cai, An & Kim, 2015*), amyloid precursor protein (*APP*) (*Selkoe & Hardy, 2016*), and apolipoprotein E (*APOE*) (*De Marco et al., 2017*) are responsible for Aβ production, has been widely accepted, and depletion of Aβ supply or obstruction of Aβ production through targeting the four genes has been a mainstay of anti-AD drug design strategy. The four genes, together with tau protein (*MAPT*) (*Hyman, 2016*) (thought to be responsible for neurofibrillary tangles, a second main hypothesis on AD pathology (*Bancher et al., 1989*)) have been termed the five AD "culprit" genes. *PENS1* and *PENS2* are parts of the γ-secretase complex, the enzyme (together with β-secretase 1, or *BACE1*) that cleaves *APP* to produce Aβ, and *APOE* enhances proteolytic breakdown of Aβ. The last few years have seen late-phase failures of the trials of many of these drugs: Semagacestat (*Bateman et al., 2009*) is an inhibitor targeting γ-secretase to obstruct Aβ production, Atabecestat (*Timmers et al., 2018*) and Verubecestat (*Egan et al., 2018*) inhibit *BACE1* (β-secretase), and Aducanumab (*Sevigny et al., 2016*), Bapineuzumab (*Tayeb et al., 2013*), Solanezumab (*Tayeb et al., 2013*) and Crenezumab (*Blaettler et al., 2016*) are humanized monoclonal antibodies designed to target Aβ. Reasons for the test failures (*Mullard, 2017*) are not known and the underlying pathophysiology of AD remains unclear.

Numerous studies have reported genetic links between AD and AG, including that AD and AG share a common set of declining synaptic genes (*Berchtold et al., 2013*), and that genes related to mitochondrial metabolism and energy production (*Wang, Michaelis & Michaelis, 2010*), and genes involved in neuronal calcium dependent signaling (*Saetre, Jazin & Emilsson, 2011*), are significantly downregulated in both AD and AG. The identification of differentially expressed genes (DEGs) (*Tusher, Tibshirani & Chu, 2002*) has been widely used in the study of complex disorders, including AD (*Avramopoulos et al., 2011*). Such studies have implicated as possible causes for AD mitochondrial and DNA damage (*Swerdlow, 2011*), inflammatory response (*Sekar et al., 2015*), ubiquitin-proteasome dysfunction (*Hong, Huang & Jiang, 2014*), and others. Recently the method of differential co-expression (DCE) analysis has been proposed as suitable for understanding biological signatures in complex diseases (*Amar, Safer & Shamir, 2013*).

Aging has long been recognized as a major risk for neurological disorders, including AD. While it is easy to tell the difference between normal aging and a state of advanced AD, differentiating between normal AG and the early onset of AD is not. The ability to detect AD at its early stages offers the best possibility of treatment, either slowing or arresting

its progress and, hopefully it the future, reversing it. In this study, our goal is to identify dysfunctional signatures of AG and AD separately, to examine how they differ, and to gain insights into recognizing signatures of early onset of AD. The materials used for this study were six sets of whole-genome gene expression microarray data, one set for AG (brain tissues from 70 years and older versus 40 years old and younger) and five sets for ALZ (tissues from five brain regions—entorhinal cortex (EC), hippocampus (HC), medial temporal gyrus (MTG), posterior cingulate (PC), superior frontal gyrus (SFG)—of 65 years and older AD patients versus age-matched healthy controls). For clarity, we use ALZ instead of AD when specifically referring to the AD datasets. From each of the six datasets (or cases) we curated sets of DEG and interacting differentially co-expressed (IDCE) genes pairs. The method of gene set enrichment and KEGG pathways were employed on these curated gene sets to identify putatively disrupted biological pathways (or functions). The enriched contents of pathways were analyzed in detail comparatively case wise. Our analysis revealed strong heterogeneity in AD signatures among the five brain regions, with HC, PC, and SFG showing clear and pronounced AD signature, MTG moderately so, and EC showing almost none. There was stark difference between ALZ and AG, the most notable being the very strong OXPHOS and Proteasome disruptions in HC/PC/SFG, but no OXPHOS disruption and only weak Proteasome disruption in AG. Our result is consistent with the Antimicrobial Protection Hypothesis of AD. We identified 54 "ALZ-only" differentially expressed genes, all down regulated and which, when used to augment the gene list of the KEGG AD pathway, made it significantly more AD-specific.

## MATERIAL AND METHODS

### Gene expression microarray data source

Six sets of gene expression microarray data (the six cases) were selected from Gene Expression Omnibus (GEO) (*Barrett et al., 2011*) (http://www.ncbi.nlm.nih.gov/geo/). (a) Five sets of ALZ data (GEO accession number GSE5281) taken from five regions of the brain selected for being known to be differentially vulnerable to the histopathological and metabolic features of AD (*Liang et al., 2007*; *Liang et al., 2008b*): entorhinal cortex (EC, from Brodmann's areas 28 and 34), hippocampus (HC), medial temporal gyrus (MTG, BA 21 and 37 and proximate BA 22), posterior cingulate (PC, BA 23 and 31), and superior frontal gyrus (SFG, mostly BA 8). Data from a sixth region, primary visual cortex (VCX; BA 17) were not included in this analysis. (b) One set of AG data (GEO project accession number GSE53890) (*Lu et al., 2014*) taken from prefrontal cortex (BA 9, 10, 11, 12, 46, and 47) of normal healthy cohort 26 to 106 years of age. From the AG data two groups were selected, the young or control group, age <40, and the old or test group, age >70. All microarray data were on the platform Affymetrix U133 plus 2.0 (Table S1).

### Database on protein-protein interaction

Protein-protein interaction (PPI) information on 12,231 human protein entries and 74,236 interactions (non-redundant) were downloaded from Human Protein Reference Database (HPRD) (*Prasad et al., 2009*) (http://www.hprd.org/) and Uniprot (*UniProt Consortium, 2014*) (http://www.uniprot.org/) and used in the construction of IDCE pairs.

### KEGG database on biological functions and pathways

The Kyoto Encyclopedia of Genes and Genomes (KEGG), a database for biological categories, including biological pathways, was used for querying the functional enrichment of gene sets (*Huang, Sherman & Lempicki, 2008*).

### Two setss of known AD target genes from AlzGene and AlzBase

Two types of known AD genes were used in this study, AlzGene (*Bertram et al., 2007*) (Alzforum; https://www.alzforum.org/) and AlzBase (*Bai et al., 2016*) (https://omictools.com/alzbase-tool). AlzGene is a collection of published Alzheimer's disease genetic association studies aimed to include GWAS meta analysis results. It contains information on 8,246 GWAS entries and 693 genes. AlzBase is a collection of genes ranked by frequency of appearance in dysregulated cellular functions in a variety of AD- and AG-related situations. For comparison with gene sets curated in this study "top" genes, called "known AD target genes" here, from the two databases were selected by using frequency thresholds: >7 for AlzGene, yielding 106 genes (the top-106 genes), and >15 for AlzBase, yielding 109 genes (the top-109 genes) (Table S2).

### Computational software

Microarray data processing and analysis were done in the R environment (http://cran.r-project.org/). Differentially expressed genes (DEGs) were selected using LIMMA (*Ritchie et al., 2015*). Microarray data were normalized using the Robust Multi-array Average (RMA) function (*Ritchie et al., 2015*) in Bioconductor (http://www.bioconductor.org/). Functional enrichment analysis of gene sets were carried out using DAVID (*Huang, Sherman & Lempicki, 2008*) (v 6.7).

### Quality screening of data and differentially expressed genes (DEGs)

A flowchart of computational procedures is sketched in Fig. 1. Some of the raw dataset had poor test-control separation. In particular, PCA analysis showed the test and control of the original SFG dataset to be substantially unseparated in component 1 (33.79%) but partially separated in component 2 (16.46%), resulting in a score of zero. Two standard quality control tests, DEG-based two-way hierarchical clustering (G2HC) and principal component analysis (PCA), were applied on the raw datasets to assure good test-control separation and similar dataset size. For each case a reduced, a "good separation" set (Table S1) of microarrays was determined by pruning microarrays from the original set until there was perfect separation in both the G2HC (Fig. S1) and PCA (Fig. S2) tests. PCA analysis of the reduced SFG dataset showed the test and control to be completely separated in component 1 (42.10%) and partially separated in component 2 (9.58%), resulting in a score of 1.0. Pruned microarrays were excluded from subsequent analyses (see effect of dataset reduction in Discussion). In the ensuing analysis in R environment, non-sense genes and gene duplications were removed from the 46,141 probe sets, leaving 21,765 genes for all cases. Significant genes were then culled according to false-discovery ratio (FDR) of the fold-change (FC) in gene expression. Approximately 800 DEGs per case were selected by LIMMA (*Ritchie et al., 2015*) with the criteria $|FC (log2)|>1$ and case-dependent FDR

**Figure 1** **Flowchart.** Microarray whole-genome gene expression datasets on aging (AG) and Alzheimer's disease cohorts (ALZ) were collected and quality-control screened; significant genes were curated using two independent methods, DEG and IDCE; six curated gene sets were compared with known AD target genes and were used to query the KEGG pathway database to identify putative disrupted biological functions in AG and ALZ; differences in disruptions in AG and ALZ were identified at the gene level to identify novel AD target genes. Results on pathway analysis were discussed and compared with literature, and inferences drawn.

thresholds (for AG, 7.50E–04; EC, 4.0E–06; HC, 1.0e−05; MTG, 5.0e−06; PC, 1.0e−04; SFG, 1.25e−05).

## Selection of differentially co-expressed gene (DCE) pairs

DCE analysis was done in programming language C. The method of Wang (*Wang et al., 2009b*) was used to evaluate the difference between test and control in co-expression correlation of gene-pairs, as follows. Separately for test and control samples, gene expressions were normalized over cohort and converted to $t$-scores. The linear regressions, $r$, of the $t$-scores were obtained for every gene-pair. A positive/negative $r$ implies the gene-pair is positively/negatively correlated (Fig. S3). A pair is said to have gain of co-expression (GOC) if $r_t > r_c$, and loss of co-expression (LOC) if $r_t < r_c$, where the subscript t (c) stands for test (control). To select gene-pairs with significant change of co-expression, the linear regression $r$ was converted to a heterogeneity statistic $Q$ (*Wang et al., 2009b*),

$$Q = (n_t - 3)(z_t - [z])^2 + (n_c - 3)(z_c - [z])^2, \tag{1}$$

where $n_t$ ($n_c$) is the test (control) cohort size, $z$ is Fisher's $z$-statistics

$$z = (1/2)\log((1 + r)/(1 - r)), \tag{2}$$

and $[z]$ is the $z$ averaged over all pairs. Gene pairs having top-0.1% Q-statistics, corresponds to a lower-bound Q-threshold ranging from 4.24 to 5.86 (Fig. S4), were selected as DCE pairs, yielding approximately 240,000 pairs for each case (Table S3).

## Construction of networks from interacting differentially co-expressed gene (IDCE) pairs

Case specific interacting gene networks (IGNs) were constructed by integration of DCE pairs with protein-protein interaction (PPI) data: the Human Protein Reference Database (HPRD) (*Prasad et al., 2009*) and Uniprot (*UniProt Consortium, 2014*). If two genes form a DCE pair and if, according to the PPI data, the pair has PPI, then the two genes form an IDCE pair and, together with the link between them, are included in the IGN. Approximately 300 genes were selected by IGNs for each case. It turned out that in each case the IGNs contained only a tiny portion of the entire set of DCE pairs. To examine the effect of this restriction to our analysis, each of the IGN was expanded to a corresponding extended IGN (xIGN) as follows: add to the IGN any gene (call it gene G) that has a PPI with at least one of the existing genes in the IGN (call it gene A) and is a DCE partner with at least another existing gene in the IGN (call it gene B), provided genes A and B are connected in the IGN. Then G and B form an IDCE pair in xIGN. By defintion A and B cannot be the same gene, otherwise G would have already been in IGN. Whereas an IDCE pair in IGN is a DCE pair with direct PPI, a newly added IDCE pair in xIGN is a DCE pair with once-removed PPI. Genes in xIGN were also required to have minimum degrees: >2 for AG and >3 for ALZ.

## Functional Profiling of the DEGs and the IGNs

Genes in each case-specific DEG, IGN, and xIGN were used separately as a gene set for querying enriched KEGG pathways using DAVID (*Huang, Sherman & Lempicki, 2008*) (v 6.7); pathways with $p$-values of Fisher's exact test less than 0.05 were considered significant and ranked by $p$-value. To simplify language, we shall say a KEGG term suffers a type 1 (putative) disruption when it is enriched in DEGs, and a type 2 disruption, when enriched in IGN (or xIGN) genes.

## RESULTS

### Properties of curated gene sets

Eighteen AG and ALZ gene sets—DEG, IGN, and xIGN sets for AG and five ALZ regions, collectively called curated gene sets, were curated in this work. With case-dependent stringent FDR thresholds about 800 DEGs were selected for each case. The ratio of downregulated to upregulated genes was about 7:3 for AG, about 5:1 for EC, HC, PC, and SFG, and about 3:4 for MTG (Table S4). A two-way clustering of the six cases based on the union of curated DEGs (containing 3,355 genes) put HC, PC, and SFG in a tightly knit group and leaves MTG, EC, and AG as outliers (Fig. S5). Because of this clustering result and because HC and PC are viewed as the most important brain regions characterizing ALZ, in this study we used HC/PC/SFG as the main group for comparison with AG. The number of genes in the interacting gene network (IGN) in each case was approximately 350. While the average degree per gene (number of links connected to it) was less than 2,

a few genes have degrees as high as 14 (Table S4). By design the extended interacting gene networks (xIGNs) were much larger than their IGN counterparts; the number of genes was controlled to be approximately 900 by admitting only genes with degree greater than 2 in the case of AG, and greater than 3 in the ALZ cases (Table S4). The top-10 most significant DEGs in the six cases had FDR values far smaller than their respective FDR thresholds (Table S5). Similarly, the highest-degree genes in the IGNs and xIGNs had degrees much higher than the threshold degree (Tables S6–S7). The overlap between the curated DEG and IGN gene sets was about 10% of IGN, except for EC (1%) and HC (24%). The overlap between IGN and xIGN was about 90% of IGN, except for PC (40%), showing that the DEG and IGN sets were substantially distinct, whereas xIGN was basically an enlargement of IGN (Fig. 2). In our discussion we focused on IGN and used xIGN only for reference.

## Curated gene sets significantly enriched in AlzBase, not AlzGene

The top-106 AlzGene and top-109 AlzBase gene sets are essentially distinct from each other, having only three genes –*PGK1*, *GAPDH*, and *CDK5* –in common. The AlzGene set, collected mainly from SNP experiments, was poorly enriched in the curated genes sets; with few exceptions the enrichment $p$-values were greater than $10^{-2}$. In contrast, the AlzBase set, collected mainly from DEG experiments, was highly enriched in the curated set; with few exceptions the enrichment $p$-values were less than $10^{-10}$, and in a majority of cases less than $10^{-15}$ (Table S8, Fig. 3). The DEG set having the highest enrichment in the AlzBase set was AG-DEG, with $p$-value = 7.5E–64, followed by SFG-DEG (1.9E–44); the set having by far the lowest enrichment was EC-DEG (3.4E–03). These observations are consistent with our two-way clustering of the six DEG sets (Fig. S5). The top-5 genes (in frequency of appearance) from AlzGene and AlzBase were distinct from each other. With one exception—the AlzGene gene *ABCA1*, an up-regulated gene in DEG sets of AG, EC, and MTG that did not appear in any of the curated IGN sets—they all appeared down-regulated genes in one or more of the curated DEG sets and in one or more of the ALZ-IGN sets. None of the top-5 AlzGene/AlzBase genes played a significant role in our (later) functional analysis of the curated gene sets (Table 1).

## Five major AD culprit genes were not prominent in the curated gene sets

The five AD culprit genes, *APP*, *MAPT* (or tau protein), *APOE*, *PSEN1*, and *PSEN2*, believed to have key roles in AD-genesis, had DCE partners in the brain regions except EC (Table S9). SFG had by far the largest number (80) of culprit related DCE pairs, with HC, at 14 pairs, a remote second. In SFG, *MAPT*, which codes the tau protein and whose over-expression can result in the self-assembly of axonal tangles, had 41 DCE partners; *APOE*, associated with the common late onset familial and sporadic forms of AD, had 24; *APP*, which codes amyloid- β protein precursor (AβPP) and whose proteolysis leads to the formation of Aβ, a primary component found in the brain of AD patients, had 10 (Table S9).

In spite of their numerous DCE partners, the culprit genes either did not appear in the curated ALZ gene sets or, if they did, were of low ranking. *PSEN1* had no IDCE partners,

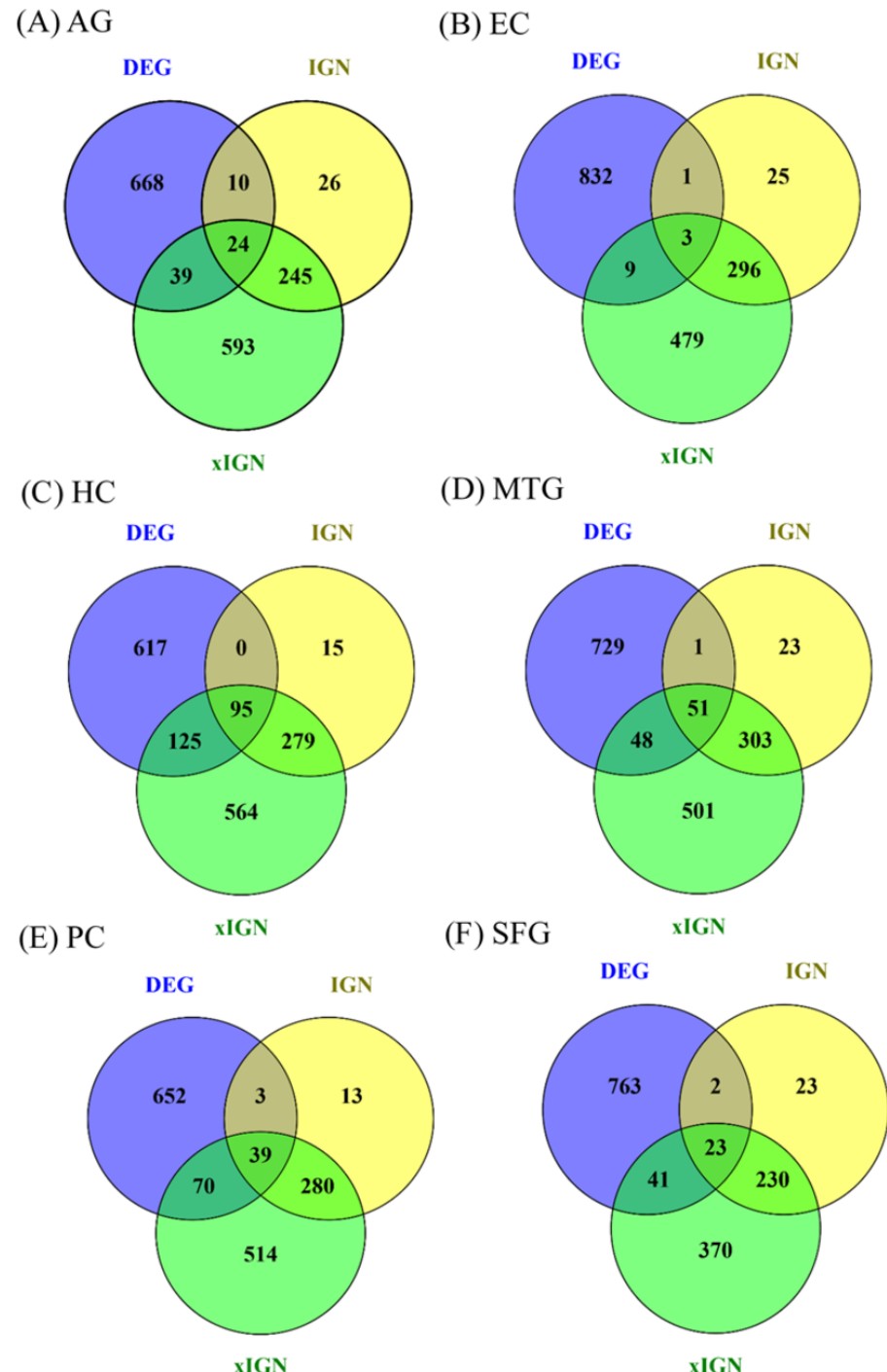

**Figure 2 Venn diagrams for six sets of DEG, IGN, and xIGN.** Vann diagrams of three curated gene sets DEG (differentially expressed gene), IGN (interacting gene network), and xIGN (extended IGN) from the six datasets: (A) old versus young, or AG, plus AD affected versus old-matched healthy brain from five brain regions, (B) entorhinal cortex (EC); (C) hippocampus (HC); (D) medial temporal gyrus (MTC); (E) posterior cingulate (PC); (F) superior frontal gyrus (SFG). Complete lists of all curated gene sets for the six cases may be accessed at the Figshare database link: 10.6084/m9.figshare.8952938.

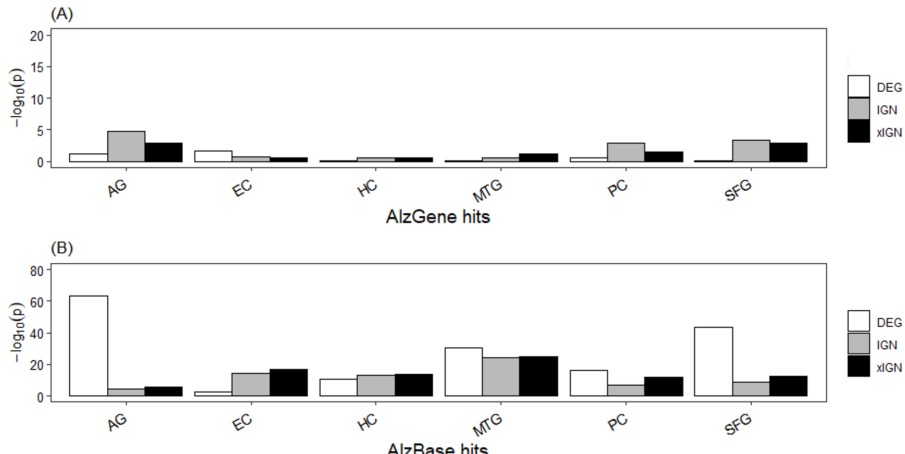

**Figure 3** **Enrichment of known AD target genes in curated AG and ALZ gene sets.** (A) Histogram, $-\log_{10}(p)$ (Fisher's exact test $p$-value) of enrichment of top-106 AlzGene genes in curated gene sets. (B) Histogram, enrichment of top-109 AlzBase genes. Abbreviation: AG, normal aging; EC, entorhinal cortex; HC, hippocampus; MTG, medial temporal gyrus; PC, posterior cingulate; SFG, superior frontal gyrus.

**Table 1** **Top-5 genes from AlzGene and AlzBase in the curated gene sets.** The top-5 genes are the five genes in each of AlzGene and AlzBase having the highest appearance frequencies. In column 2 (DEG) and 3 (IGN), datasets in which the gene occurs (when it does) are given. Number in brackets gives log2 fold change (down-regulated if negative) in the case of DEG and degree (only genes with degree 2 or greater are included). None of the top-5 genes appear in any of the gene lists of the three KEGG pathways hsa05130: *E. coli* infection, hsa00190: oxidative phosphorylation (OX-PHOS), and hsa03050: Proteasome.

| Gene symbol | Curated AG and ALZ gene sets | | Ref. |
|---|---|---|---|
| | **AlzBase** | | |
| | DEG | IGN | |
| NSF | AG (−1.31), MTG (−3.20), SFG (−2.88) | EC (4), HC (3), MTG (10), SFG (2) | *Hou et al. (2017)* |
| NECAP1 | AG (−1.13), HC (−2.36), PC (−2.46), SFG (−2.75) | AG (2), EC (2), HC (3) | *Okamoto et al. (2018)* |
| MDH1 | AG (−1.25), MTG (−3.72), PC (−2.96), SFG (−3.14) | EC (3), HC (6), MTG (4) | *Sonntag et al. (2017)* |
| AMPH | AG (−1.31), MTG (−3.43), SFG (−2.65) | AG (3), EC (3), MTG (4) | *Xu et al. (2019)* |
| [a]PGK1 | AG (−1.22), HC (−2.71), PC (−2.40) | EC (2), HC (5), MTG (2) | *Labudova et al. (1999)* |
| | **AlzGene** | | |
| [a]GAPDH | HC (−3.88), PC (−3.39) | HC (5), MTG (4), PC (6) | *El Kadmiri et al. (2014)* |
| UBQLN1 | AG (−1.17) | AG (3), EC (5), HC (4) | *Li et al. (2017)* |
| ABCA1 | AG (1.32), EC (1.61), MTG (2.21) | – | *Nordestgaard et al. (2015)* |
| [b]CDK5 | EC (−2.20), SFG (−1.80) | PC (2) | *Liu et al. (2016)* |
| GSK3B | HC (−1.75) | AG (2), MTG (5) | *Llorens-Marítin et al. (2014)* |

**Notes.**
EC, entorhinal cortex; HC, hippocampus;; MTG, medial temporal gyrus; PC, posterior cingulate; SFG, superior frontal gyrus.
[a]Genes were also used as housekeeping genes in AD studies
[b]Gene appears in both AlzGene and AlzBase sets

either in IGN or xIGN. *APOE* was the only culprit gene that appeared in IGN; it had a single IDCE partner in SFG. The culprit genes had more IDCE partners in the larger xIGNs. Two culprit genes were among the DEGs: *PSEN1* in EC (upregulated) and PC (downregulated), and *PSEN2* in MTG (downregulated). *MAPT* and *APOE* were in both the curated IGN and xIGN sets of AG (Table S10).

### KEGG pathways were enriched heterogeneously in curated gene sets

Many KEGG pathways were heterogeneously enriched (Fisher's exact test *p*-values <0.05) in the curated gene sets. Relative to the size of the curated set, more KEGG pathways were enriched in the IGN sets than in the DEG sets. This probably is because the IGNs were distilled interaction networks; interrelation between genes in the IGNs was more similar to that in the KEGG pathways (Table S11). The patterns of KEGG pathways enriched in curated IGN and xIGN sets were broadly similar, but were substantially different from those enriched in curated DEG sets (Tables S12–S14). In what follows we shall focus on the DEG and IGN sets.

The categories of KEGG pathways enriched in DEG and IGN genes for AG and ALZ regions were substantially different, despite important similarities (Tables S12–S14). Category of pathways both DEG- and IGN-enriched included cell proliferation, neurodegeneration, protein complexes and metabolism, and inflammation. Pathways in the category of metabolic related pathways were all DEG-enriched, but not IGN-enriched, and those in carcinoma, the opposite (Fig. 4).

### OXPHOS and pathways of the three neurodegenerative diseases highly enriched in DEG sets of HC, PC, and SFG

The most prominent feature of the KEGG analysis of the DEGs were the extremely high enrichment of (the KEGG pathways) Oxidative phosphorylation (OXPHOS), Parkinson's disease (PD), Hunting's disease (HD) and Alzheimer's disease (AD) in the DEG sets of three regions HC, PC, and SFG. In addition to these four pathways, Proteasome, Pyruvate metabolism, and TCA cycle were also among the Top-10 enriched KEGG pathways in the three regions, and *Vibrio cholerae* infection and Pathogenic *E. coli* infection, in two of the three regions. KEGG enrichment in the other three cases were weaker; in order of descending enrichment significance: AG, MTG, and EC. Top-10 enriched pathways in MTG include OXPHOS, *V. cholerae* infection, and Pathogenic *E. coli* infection, and Epithelial cell signaling in *H. pylori* infection, which it shared with AG and SFG. This last pathway was the only top-10 KEGG pathway AG had common with the ALZ cases. EC, which was identified as an outlier in our earlier comparative study of the curated DEG lists, had only four significantly enriched KEGG pathways, including HD (Fig. 4, Table S12).

### Proteasome was the KEGG pathway by far the most enriched in IGN genes

Proteasome was by far the most prominent pathways in the KEGG analysis of curated IGN genes; it was the pathway with the highest enrichment significance in five cases including AG, but was not significant in SFG. Pathogenic *E. coli* infection significant was significant in all six cases. Ribosome, not seen in DEG analysis, was significant in all five ALZ cases.

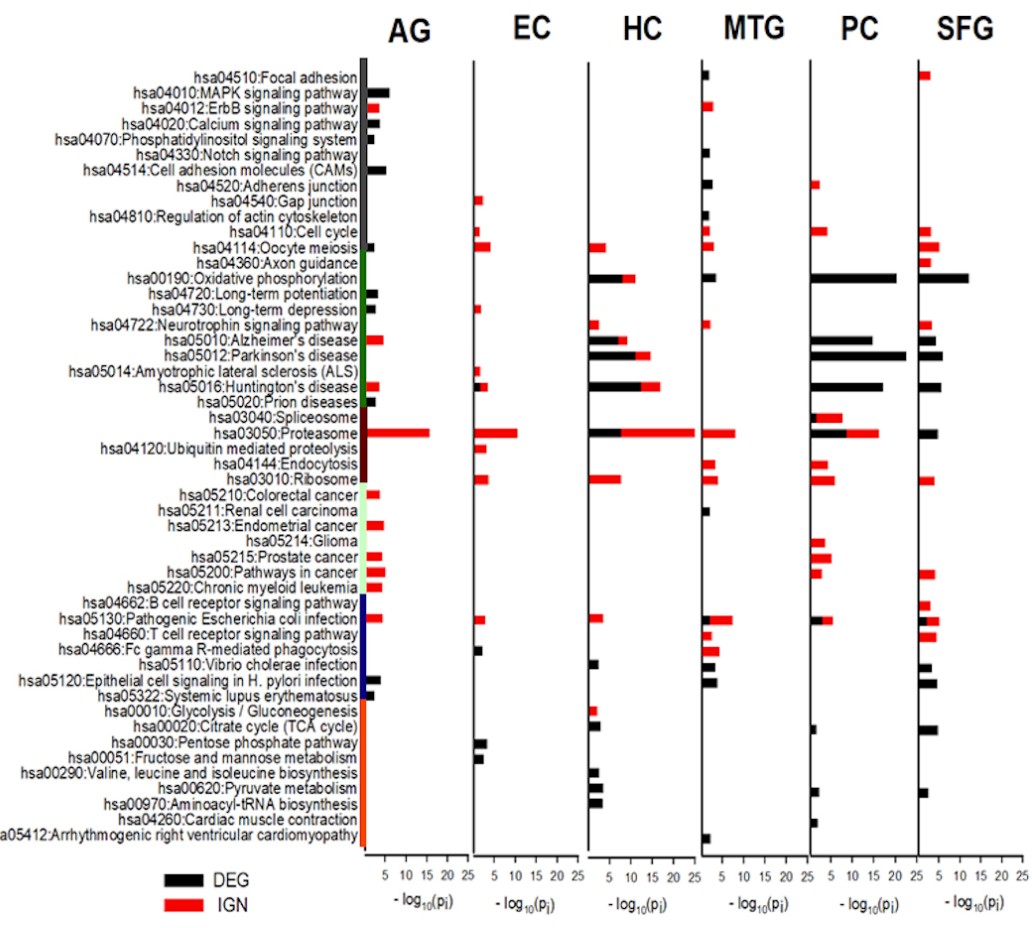

**Figure 4** **KEGG pathways most enriched in curated DEG and IGN gene sets.** The six columns of horizontal histograms give $-\log_{10}$ p-values of top-10 significantly enriched KEGG pathways in the six case: from left, AG, normal aging; EC, entorhinal cortex; HC, hippocampus; MTG, medial temporal gyrus; PC, posterior cingulate; SFG, superior frontal gyrus. Black (red) towers are for DEG (IGN) enrichment. Categories of KEGG pathways are indicated by color bars to the left of the histogram plot: dark gray, cell proliferation; green, neuro-degeneration; brown, protein complexes and metabolism; cyan, carcinoma; dark blue, inflammation; orange, metabolic related pathways. See Tables S12–S13 for further details of enriched KEGG pathways in DEG, IGN, and xIGN gene sets.

OXPHOS, AD, PD, HD were significant in HC (but less so than in DEG) but not in PC and SFG, nor in MTG. AD and HD was also significant in AG, and HD (weakly) in EC. Five cancer pathways, Pathways in cancer, Endometrial cancer, Prostate cancer, Chronic myeloid leukemia, Colorectal cancer were significant in AG, the last three pathways also in PC, and last pathway in PC. No cancer related pathways were significant in HC, MTG, and EC. (Fig. 4, Table S13).

## DEG and IGN genes enriched in the same KEGG pathway tend to be distinct

When a KEGG pathway was both DEG- and IGN-enriched in a given case, the enriched genes from the two lists were generally quite different, perhaps not surprising because

selection criteria for DEG—significant change in expression level—and IGN –significant change in correlation of expressions of gene pair—were distinct, and because only a small portion of genes are common to both DEG and IGN (Fig. 2). Enriched genes present in both IGN lists of AG and ALZ tended to have very different GOC and LOC linkages, and some enriched genes present in multiple curated ALZ gene lists were absent in the corresponding AG lists.

### ALZ-only hits and novel target genes for the KEGG AD pathway

Excluding the three neurodegenerative diseases, pathways that were among at least two of the three top-10 KEGG pathways of HC/PC/SFG were OXPHOS, Proteasome, Pyruvate metabolism, TCA cycle, *V. cholerae* infection and Pathogenic *E. coli* infection (Tables S15–S16). Genes in the gene lists of these six pathways that were DEG hits in at least two of HC/PC/SFG but not in AG were collected into a set of 54 "ALZ-only hits", of which 43, 47, and 36, respectively, were DEGs of HC, PC, and SFG; all were down-regulated (Fig. 5 and Table S16). AlzBase contains two of the hits, *ATP5B* and *ATP5G3*, and AlzGene, none. Of the 54 hits, 26 were from OXPHOS, 14 from Proteasome, and 10 from Pyruvate metabolism and TCA cycle combined (one shared with OXPHOS and 4 are common to both), and 5 from *E. coli* infection. All OXPHOS hits except *SDHA* (the hit shared with TCA cycle) were mitochondrial enzymes: 9 belong to the ATP family of genes; 9, NDU; 4, COX; and 3, UQCR. None of the hits not in OXPHOS were mitochondrial. All Proteasome hits except POMP were from the PSM family of enzymes. All the OXPHOS hits except *ATP6V1E1*, *ATP6V1H*, *ATP5J2*, and *ATP5L* were also in the KEGG AD pathway, and none of the other 32 hits, call "novel AD genes", were (Table 2, Fig. 5).

## DISCUSSION

### Comparison with original analysis of AG dataset

In a comprehensive study, the authors of the AG dataset presented evidence suggesting that the transcription factor *REST* (also known as neuron-restrictive silencer factor, NRSF) is normally induced and acts as a universal feature of normal ageing in human cortical and hippocampal neurons, and a neuroprotective modulator by repressing genes that promote cell death and the pathology of AD, and confers oxidative stress resistance and protects against toxic insults, such as Aβ oligomers and tau phosphorylation, associated with AD, but is lost in mild cognitive impairment and AD (*Lu et al., 2014*). In our study the gene *REST* was moderately up-regulated in AG and, in spite of what was suggested in (*Lu et al., 2014*), similarly in all five ALZ datasets, but, with a FC (log2) ranging from 0.541 (PC) to 0.914 (SFG), was not a DEG (Table S17). Our result does not directly contradict (*Lu et al., 2014*) because the AD work in *Lu et al. (2014)* was not genomic based. Our ALZ study did yield many strong AD signatures, including a severe dysfunction of OXPHOS.

### Comparison with original analyses of ALZ datasets

The ALZ datasets (we did not analyze the dataset collected from the primary visual cortex) were also analyzed in *Liang et al. (2007)*, *Liang et al. (2008b)* and *Liang et al. (2008a)*. In methodology the main difference between the analyses of these authors and us was, in our

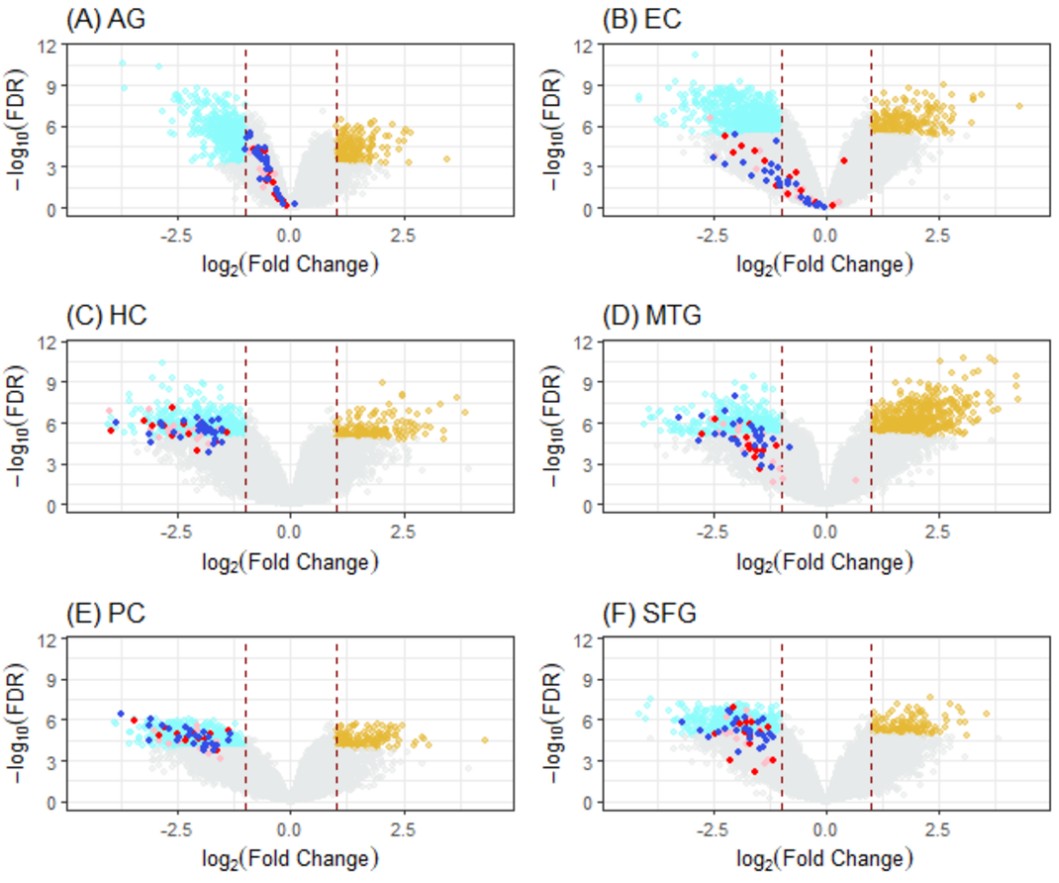

**Figure 5  Volcano plots of ALZ-only hits from six DEG cases.** Cases (A) to (F) correspond to volcano plots for the six DEG sets: AG, normal aging; EC, entorhinal cortex; HC, hippocampus; MTG, medial temporal gyrus; PC, posterior cingulate; SFG, superior frontal gyrus. DEGs (cyan, downregulated; light brown, upregulated) were selected using LIMMA with case dependent FDR (for AG, 7.50E−04; EC, 4.0E−06; HC, 1.0e−05; MTG, 5.0e−06; PC, 1.0e−04; SFG, 1.25e−05) and |FC(log2)|>1 (red dashline). Dots are "ALZ-only" hits (Table 2) in the gene list of leading KEGG pathways enriched in DEGs, including OXPHOS (deep-blue), Proteasome (red), and others (pink).

case, the reduction of the datasets so that all datasets were of similar size (about 10 vs. 10) and, more importantly, had good test-control separation in PCA analysis. In addition, for function analysis *Liang et al. (2008a)* used Gene Ontology (GO) and we used KEGG pathways. In practice our DEG results are quantitatively (Table S18) and qualitatively consistent with the results in (*Liang et al., 2008a*), except for the case of SFG. This is perhaps to be expected when the PCA analysis for the original SFG set (23 control vs. 11 test) had a score of zero, whereas the reduced set (12 vs. 8) had a score of 1. When the screening criteria (FDR <1.25E–05, |FC|>1) used to selected the 829 DEGs from the reduced SFG set were applied to the original set the yield was zero; using the criteria FDR <1.00E–02, |FC|>1 yielded 395 DEGs, which had 92 common genes with the 829-gene set (Table S18). The functional analyses also yielded drastically different results for SFG. In our result SFG belonged to the same category as HC and PC, whereas in *Liang et al.*

*(2008a)* it was an outlier and shared no common feature with HC or PC. For instance, mitochondrial functions were reported to be severely disrupted in HC, PC, and MTG but not in SFG (*Liang et al., 2008b*), whereas our result indicated it to be severely disrupted in all four regions. In both analyses EC was also an outlier, separate from SFG in (*Liang et al., 2008a*). We emphasize that concerning SFG there is no contradiction between ours and the analysis in *Liang et al. (2007)*, *Liang et al. (2008b)* and *Liang et al. (2008a)*; different sets of microarrays were analyzed, which yielded different presentations of the pathology of SFG, differences which may be resolved by new data and analyses.

## Comparison with AlzGene and AlzBase

That our curated gene set has very low enrichment in the AlzGene set but much higher enrichment in the AlzBase set (Fig. 3) could be because gene expression was used to selected genes in AlzBase, as was in the present work, whereas SNP, one of the most common types of genomic variation, was used to selected genes in AlzGene. The AlzBase set was more enriched in our AG than ALZ gene sets because it (AlzBase) was curated from AD brain as well as AG brain (*Bai et al., 2016*). Gene expression as a leading determinant factor may explain why the DEG subsets of the curated genes were, overall, more enriched than the IGN and xIGN subsets in the AlzBase set, which was essentially determined by DEG. The IGN and xIGN subsets were determined by co-expression of gene-pairs as well as PPI data. It is notable that the five major AD culprit genes were not prominent among the curated genes. In comparison, the top-106 AlzGene gene set contains the four culprit genes *APOE, MAPT, PSEN1*, *APP* (in descending order of appearance frequency), but the top-109 AlzBase gene set contains none. Culprit genes most likely were more active during the onset of AD in the patients and less, or even not, active during late stages of AD. In contrast, curated genes were culled from microarray data taken postmortem from patients, whose deaths presumably were caused by advanced states of AD.

## Cancer-related pathways disrupted in AG, essentially not in ALZ

Most of the leading disrupted KEGG pathways in AG were not disrupted in ALZ. Those that were type 1 disrupted were mainly signaling pathways related to cancer: MAPK signaling pathway (*Dhillon et al., 2007*), Cell adhesion molecules (*Bendas & Borsig, 2012*), and Calcium signaling pathway (*Monteith, Prevarskaya & Roberts-Thomson, 2017*) (Table S12), and type 2 disrupted were pathways related to specific cancers: Pathways in cancer, Endometrial cancer, Prostate cancer, and Colorectal cancer (Table S13). Because AG compared old (70 years and above) versus young (40 and below), whereas ALZ compared AD patients versus age-matched (65 and above) non-AD cohorts, the above results do not imply AD patients did not have cancer-related dysfunctions, rather, it meant such dysfunctions were not exacerbated by AD. It has been suggested that an inverse comorbidity relation exists between AD and some types of cancer, including smoke-related ones (*Driver et al., 2012*; *Musicco et al., 2013*). Our result does not directly support it.

## OXPHOS and Proteasome strongly disrupted in ALZ, not in AG

Brain related dysfunctions, especially mitochondrial dysfunction, are often discussed in the context of both old age and AD. In this work there was clear distinction between the two
**Table 2** **Fifty-four AD-only hits and 32 novel AD genes.** The 54 AD-only hits were enrichment hits from 6 KEGG pathways—OXPHOS (Oxp), Proteasome (PSM), Pyruvate metabolism (Pym), TCA cycle (TCA), *V. cholerae* infection (Vch) and Pathogenic *E. coli* infection (Ecol)—in two of three DEG sets of HC/PC/SFG and not in AG. The 32 genes not in the KEGG AD pathway (AD) are called ''novel AD genes''. Gene families: acylphosphatase (ACYP), Actin Related Protein 2/3 Complex (ARPC), adenosine triphosphate (ATP), cyclooxygenase (COX), -ketoglutarate dehydrogenase (DLD), fumarate hydratase (FH), glyoxalase (GLO), isocitrate dehydrogenase (IDH), lactate dehydrogenase (LDH), malate dehydrogenase (MDH), NADH:ubiquinone oxidoreductase (NDU), pyruvate dehydrogenase (PDH), proteasome maturation protein (POMP), proteasome subunits (PSM), succinate dehydrogenase (SDH), ubiquinol-cytochrome c reductase (UQCR). Abbreviations for the six cases: EC, entorhinal cortex; HC, hippocampus; MTG, medial temporal gyrus; PC, posterior cingulate; SFG, superior frontal gyrus.

| KEGG | Protein families | Gene | DEG | Up or Down | IGN | Reference & remark |
|---|---|---|---|---|---|---|
| | ARPC | *ARPC1A* | HC, PC, SFG | Down | | Arp2/3 complex is a central player in actin-based motility of pathogens. |
| | | *TUBA1B* | HC, PC, SFG, MTG | Down | AG, EC, HC, MTG, PC, SFG | |
| | | *TUBA1C* | HC, PC, MTG | Down | AG, HC, MTG, PC, SFG | |
| Ecol | Tubulin | *TUBB* | HC, PC, SFG | Down | AG, EC, HC, MTG, PC, SFG | Level of E. coli K99 was reported to be greater in AD compared to control brains(*Zhan et al., 2016*). Reduced $\alpha$-tubulin expression led to increased human-tau expression in transgenic worm (*Miyasaka et al., 2018*). |
| | | *TUBB3* | PC, SFG, MTG | Down | AG, EC, MTG | |
| | ACYP | *ACYP2* | HC, PC | Down | | |
| Pym | GLO | *GLO1* | HC, PC, MTG | Down | | |
| | LDH | *LDHB* | HC, PC | Down | | Decreased LDHB relative to LDHA leads to CNS aging in transgenic mice (*Ross et al., 2010*). |
| | DLD | *DLD* | HC, PC, SFG | Down | HC, PC | DLD activity down in AD-human brain (*Bubber et al., 2005*). |
| | MDH | *MDH2* | HC, SFG | Down | EC | MDH activity up in AD-human brain (*Bubber et al., 2005*). |
| Pym, TCA | PDH | *PDHA1* | HC, SFG | Down | | PDH activity down in AD-human brain (*Bubber et al., 2005*) and transgenic mice (*Yao et al., 2009*). |
| | | *PDHB* | PC, SFG | Down | HC, SFG | |
| | FH | *FH* | PC, SFG, MTG | Down | | Activity changed in AD-human brain (*Bubber et al., 2005*). |
| TCA | IDH | *IDH3G* | EC, HC, SFG | Down | | IDH activity down in AD-human brain (*Bubber et al., 2005*). |
| TCA, Oxp, AD | SDH | *SDHA* | HC, PC | Down | AG | SDH activity down in transgenic mice (*Yao et al., 2009*). |
| Vch, Oxp | ATP | *ATP6V1E1* | HC, PC, SFG, MTG | Down | EC, HC, MTG | |
| | | *ATP6V1H* | HC, PC | Down | EC, HC, MTG | |

Peerj

**Table 2** (*continued*)

| KEGG | Protein families | Gene | DEG | Up or Down | IGN | Reference & remark |
|------|------------------|------|-----|------------|-----|---------------------|
| Oxp | ATP | *ATP5J2* | PC, SFG, EC, MTG | Down | | |
| | | *ATP5L* | PC, SFG, MTG | Down | | |
| | | *ATP5A1* | HC, PC | Down | EC, HC, SFG | |
| | | *ATP5B* | HC, PC, SFG | Down | EC, HC, MTG, SFG | |
| | ATP | *ATP5C1* | PC, SFG, MTG | Down | EC, HC, MTG, SFG | ATP activity down in transgenic mice (*David et al., 2005*). |
| | | *ATP5G3* | HC, PC, SFG | Down | | |
| | | *ATP5O* | HC, PC, SFG | Down | EC | |
| | COX | *COX4I1* | HC, PC | Down | | |
| | | *COX5B* | HC, PC | Down | | |
| | | *COX6B1* | HC, PC, MTG | Down | | COX activity down in transgenic mice (*David et al., 2005*; *Yao et al., 2009*). |
| | | *COX6C* | HC, PC | Down | | |
| | | *NDUFA2* | PC, SFG | Down | | |
| | | *NDUFA8* | HC, SFG, MTG | Down | | |
| | | *NDUFA9* | HC, PC, SFG | Down | HC | |
| | | *NDUFAB1* | HC, PC, SFG, MTG | Down | | |
| Oxp, AD | | *NDUFB10* | HC, PC | Down | | |
| | NDU | *NDUFC2* | PC, SFG | Down | | NDU activity down in transgenic mice (*David et al., 2005*; *Rhein et al., 2009*). |
| | | *NDUFS3* | HC, PC, SFG | Down | PC | |
| | | *NDUFS5* | HC, PC, SFG, MTG | Down | | |
| | | *NDUFV2* | HC, PC, SFG | Down | HC | |
| | | *UQCRC2* | HC, PC, SFG | Down | AG | |
| | UQCR | *UQCR10* | HC, SFG | Down | | |
| | | *UQCRH* | HC, SFG, MTG | Down | AG, HC, PC | |
**Table 2** (*continued*)

| KEGG | Protein families | Gene | DEG | Up or Down | IGN | Reference & remark |
|---|---|---|---|---|---|---|
| | | *POMP* | HC, PC | Down | EC, HC, PC | Overexpression of POMP enhances antioxidant defense (*Chondrogianni & Gonos, 2007*). |
| | | *PSMA1* | HC, PC, SFG, MTG | Down | AG, EC, HC, MTG, PC | |
| | | *PSMB1* | HC, PC | Down | AG, MTG | |
| | | *PSMB2* | HC, SFG | Down | HC | |
| | | *PSMB3* | HC, PC, SFG | Down | HC | |
| | | *PSMB4* | HC, PC | Down | | |
| | | *PSMB5* | PC, SFG | Down | HC, MTG | |
| | | *PSMB6* | HC, PC | Down | | |
| | | *PSMC1* | HC, PC, SFG | Down | AG, HC | |
| | | *PSMC2* | HC, PC, SFG | Down | AG, EC, HC, MTG, PC | In AD-affected brain, PSMB1/6 (chymotrypsin-like) and PSMB5 (caspase-like) mediated activities decreased, but no change* in level of α- and β-subunits (*Keller, Huang & Markesbery, 2000*). Remark: *no significance assigned to measure. |
| PSM | PSM | *PSMC5* | HC, PC, MTG | Down | AG, EC, MTG, SFG | |
| | | *PSMD4* | HC, PC | Down | AG, EC, HC, MTG, PC, SFG | Aβ inhibits chymotrypsin-like (PSMB5) activity of proteasome (*Checler et al., 2000*). |
| | | *PSMD8* | PC, SFG | Down | HC, MTG, PC | |
| | | *PSMD12* | HC, PC | Down | EC, HC | |

conditions. OXPHOS and Proteasome were the two most strongly disrupted pathways in ALZ –40 of the 54 ALZ-only genes were either mitochondrial or proteasomal, yet OXPHOS was not at all disrupted, and Proteasome was only type 2 disrupted, in AG (Tables S12 and S13). The Proteasome genes enriched in the IGN-AG gene list were mostly distinct from those enriched in IGN-ALZ list (Table 2). This suggests loss of mitochondrial and/or proteasomal function in the non-AD aged has the potential of being used as an indicator for early onset of AD.

## OXPHOS and the three neurodegenerative pathways

The four KEGG mitochondria related pathways OXPHOS, AD, HD, and PD, together with Proteasome, exhibited by far the strongest type 1 and type 2 disruptions in the ALZ datasets, especially in HC/PC/SFG (a KEGG pathway is type 1 or 2 disrupted when it is enriched in a DEG or IGN set). Of the 133 genes in (the) OXPHOS (gene set) only 6 are not from one of the six families of (genes coding for) mitochondrial enzymes: NADH dehydrogenase (NDH), NADH: ubiquinone oxidoreductase (NDU), succinate dehydrogenase complex (SDH), cytochrome c oxidase (COX), ATP synthase (ATP), ubiquinol-cytochrome c reductase (UQCR). Mitochondrial genes are also preponderant in the gene lists of the three neurodegenerative diseases: 89/171, 95/142, and 89/193, respectively, in AD, PD, and HD. The DEGs of HC/PC/SFG were also rich in mitochondrial genes of OXPHOS, 28, 42, and 32 genes, respectively, in HC, PC, and SFG (Table S19). Aside from the AD culprit genes related to Aβ production, the 68 AD-specific genes (not common with OXPHOS or HD or PD, hence also non-mitochondrial) in AD are dominated by genes related to signal transduction, including genes in the CAC (Calcium Voltage-Gated Channel), CAL (Calmodulin), CAP (Calpain, Calcium-Activated Neutral Proteinase), GRIN (Glutamate Ionotropic Receptor NMDA), ITPR (Inositol 1,4,5-Trisphosphate Receptor). Only 7 of these 68 AD-specific genes were hits in DEG-HC (6/5 hits in PC/SFG), less than the 8 hits in DEG-AG. In comparison, 6/19/14/10 of the HD and/or PD-specific genes were hits in AG/HC/PC/SFG, suggesting that the KEGG AD pathway, as it currently stands, is not particularly AD specific (Table S19).

## OXPHOS dysfunction and tau and Aβ pathologies

OXPHOS is the metabolic pathway in the mitochondria that produces ATP, whose energy is released to power the brain. Dysfunction of mitochondria has long been associated with AD (*Shoffner, 1997*; *David et al., 2005*; *Phillips, Simpkins & Roby, 2014*; *Lin & Beal, 2002*), PD (*Winklhofer & Haass, 2010*), and HD (*Damiano et al., 2010*). Experiments and functional analysis on P301L tau transgenic mice demonstrated mitochondrial dysfunction leads to reduced NDU activity and impaired ATP and the suggestion that tau pathology involves a mitochondrial and oxidative stress disorder possibly distinct from that caused by Aβ synthesis (*David et al., 2005*; *Eckert et al., 2010*). Experiments on triple transgenic AD mice found a massive deregulation of 24 proteins, including 3 in the NDU family (in mitochondria complex I) and 3 in the COX family (complex IV). Because deregulation of complex I was tau dependent, whereas deregulation of complex IV was Aβ dependent, the authors concluded that OXPHOS was synergistically impaired by Aβ and tau (*Eckert*

*et al., 2010*; *Rhein et al., 2009*). Results in this work suggest wide spread mitochondrial dysfunction; the ALZ-only list includes 9 *NDU*, 1 *SDH* (complex II), 4 UQCR (complex III), 4 COX, and 9 ATP (complex V) coding genes (Table 2, Fig. 5), all down-regulated; but not any ND (complex 1) coding genes.

## The Proteasome and Aβ pathology

The ubiquitin-proteasome system (UPS) is an important part of the proteolysis machinery in eukaryotic cells; it maintains proteostasis (*Lecker, Goldberg & Mitch, 2006*), regulates proteins biosynthesis, and controls levels of presynaptic protein (*Speese et al., 2003*). UPS was associated with AD when ubiquitin was detected in neurofibrillary tangles and senile plaques in AD affected brains (*Perry et al., 1987*; *Mori, Kondo & Ihara, 1987*). Proteasome is the core of UPS that degrades ubiquitinated proteins. Cell model experiments have shown that the proteasome is inhibited by, but does not directly degrade Aβ. Rather, it degrades PS1 and PS2 as well as their mutated forms. Mutated *PSEN1/PSEN2* leads to the production of the plaque-forming peptide, Aβ42, and the inhibition of proteasome was shown to enhance this production (*Checler et al., 2000*). In the proteasome, degradation of ubiquitinated proteins/peptides is carried out in the barrel-shaped 20S complex, composed of four types—α/β/γ/δ –of proteasome subunits (*PSMA/B/C/D*). The β-subunits contain peptide cleaving activities, caspase-like by *PSMB1*/6, trypsin-like by *PSMB2/4/7*, and chymotrypsin-like by *PSMB5*, whereas the other three type of subunits have structural and regulatory functions. Experiments on autopsied AD affected brains (and controls) recorded a decrease in caspase-like and chymotrypsin-like activities in AD brain, but saw no significant decrease in either *α*- or β-subunit expression (*Keller, Hanni & Markesbery, 2000*). However, whereas the authors reported statistical significance for the activity results, the same was not provided for the expression result. The gene *POMP* encodes a molecular chaperone that is essential for 20S proteasome formation. Its overexpression increases proteasome function and enhances proteasome-mediated antioxidant defense (*Chondrogianni & Gonos, 2007*). Strong type 1 and type 2 disruptions of Proteasome was one of the most prominent features of our result. In contrast the only ubiquitin related pathway noticeably enriched in our data was hsa04120: Ubiquintin mediated proteolysis, in IGN-EC. The ALZ-only hits from Proteasome include *POMP* and 13 PSM genes covering all four types of subunits, including six of the seven β-subunits, *PSMB1* to *PSMB6*; all were down regulated (Table 2, Fig. 5). The KEGG pathway hsa04142: Lysosome pathway, system of organelles that digests waste macromolecules autophagy in the cytoplasm, did not surface in our work.

## Pyruvate metabolism, TCA cycle, and dementia

In adult human, the brain accounts for about 2% of the body weight but consumes about 25% of body glucose. OXPHOS, pyruvate metabolism, and the TCA (Krebs) cycle are key glucose metabolic pathways. Pyruvate, the end product of glycolysis, is converted by the pyruvate dehydrogenase (PDH) complex to acetyl CoA, which is taken by the TCA cycle to produce nicotinamide adenine dinucleotide (NADH), which is fed into the OXPHOS process to produce ATP. Aberrant glucose metabolism is a feature of AD

pathology (*Chen & Zhong, 2013*). In addition to PDH, key enzymes in pyruvate metabolism and the TCA cycle include lactatedehydrogenase (LDH), $\alpha$-ketoglutarate dehydrogenase (DLD), isocitrate dehydrogenase (IDH), malate dehydrogenase (MDH), and succinate dehydrogenase (SDH). A reduced glycolytic energy production, such as caused by aberrant pyruvate metabolism, is a common symptom of AD patients (*Gray, Tompkins & Taylor, 2014*). Decreased PDH and COX activity in female 3xTg-AD mice have been reported (*Yao et al., 2009*). Activities of TCA cycle enzymes in brains from patients with autopsy-confirmed AD and clinical dementia ratings (CDRs) before death have been measured, and significant ($p < 0.01$) decreases in the activities of the PDH complex, IDH, and the DLD complex, and increases in SDH and MDH were reported (*Bubber et al., 2005*). Experiments on prematurely aging mtDNA mutator mice suggested that decreased LDHB (relative to LDHA) leads to CNS aging in mice (*Ross et al., 2010*). In this work both Pyruvate metabolism and TCA cycle pathways were type 1 disrupted in HC/PC/SFG ($p$-value = 2.9E–04/6.1E–03/3.4E–03 for Pyruvate; 1.4E–03/2.7E–02/1.4E–05 for TCA), and neither was in AG. Seven enzyme coding genes associated with the two pathways, including those encoding *LDHB*, *DLD*, *MDH2*, *PDHA1*, *PDHB*, were among the ALZ-only genes; all were down-regulated (Table 2, Fig. 5).

## Pathogenic *E. coli* infection, tubulin, and tau tangle

Microtubules are dynamic structures that form part of the cytoskeleton and are composed of heterodimers of $\alpha$- (TUBA) and β-tubulin (TUBB). The aggregation of the microtubule-associated protein tau in the form of tangles in the brain is one of the two (the other being aggregation of Aβ plaques) manifests of AD (*KoSIK, Joachim & Selkoe, 1986*). Bacterial pathogens including *E. coli* (and *V. cholerae*) induce microtubule destruction in their invasion of host (*Radhakrishnan & Splitter, 2012*). Level of *E. coli* K99 was reported to be greater in AD compared to control brains (*Zhan et al., 2016*). Loss of microtubule (*Cash et al., 2003*) and tubulin (*Zhang et al., 2015*) have been reported in the neurons of AD brains. Level of reduced $\alpha$-tubulin expression led to increased human-tau expression in transgenic worm (*Miyasaka et al., 2018*). In this work the KEGG pathway Pathogenic *E. coli* infection was type 1 disrupted in HC/PC/SFG ($p$-value = 3.9E–02/1.1E–03/7.2E–03) and MTG (7.2E–03), as well as in AG (6.7E–03), and type 2 disrupted in all six cases. The ALZ-only set had five genes associated with the pathway, including two $\alpha$-tubulin genes *TUBA1B* and *TUBA1C* and two $\beta$-tubulin genes *TUBB* and *TUBB3*; all were down-regulated (Table 2, Fig. 5).

*V. cholerae* is gram-negative bacterium known for causing the disease cholera. The KEGG pathway *V. cholerae* infection was type 1 disrupted in HC, SFG, MTG as well as in AG. Of this KEGG pathway, the 14 ATPase-coding hits in AG and/or ALZ were also OXPHOS hits in AG/ALZ, two—*ATP6V1E1* and *ATP6V1H*—were ALZ-only; both are already in the gene list of the AD pathway. There were a few non-ATPase-coding hits but none satisfied ALZ-only criteria. No literature was found specifically linking *V. cholerae* to AD.

## Antimicrobial protection hypothesis of AD

Recently it has been shown, based on experiments in which Aβ-expressing, APP knockout, and mouse models, as well as worm and mammalian cell models were infected with pathogens including *C. albicans* and *S.* Typhimurium, that Aβ may be a natural antibiotic that protects the brain from bacterial infection through the adhesion to and agglutination of invading microbes by oligomerized and fibrillized Aβ (*Kumar et al., 2016*). This work led to the proposal of a new AD amyloidogenesis model, in which Aβ deposition is an early innate immune response to microbial invasion, during which Aβ first entraps and neutralizes invading pathogens, then fibrillizes and drives neuroinflammatory pathways that help fight the infection and clear Aβ and pathogen deposits. Chronic activation of these pathway leads to sustained inflammation and AD (*Moir, Lathe & Tanzi, 2018*). The new model, called the Antimicrobial Protection Hypothesis of AD by its authors, retains the Aβ production mechanism of the Amyloid Cascade Hypothesis. Implicit in the new hypothesis is AD tends to be associated with old age because the mechanism for Aβ clearance gradually breaks down with age. The present work is consistent with the new hypothesis in that three pathogens infection related KEGG pathways showed disruption in our datasets: Epithelial cell signaling in *Helicobacter pylori* infection, type 1 in AG, MTG, and SFG; Pathogenic *E. coli* infection, type 1 in all cases except EC, and type 2 in all six cases; *Vibrio cholerae* infection, type 1 in AG, HC, MTG, and SFG (Fig. 4, Table S12). Recently, experiments in transgenic mouse model have shown that acute treatment of Aβ induces microglia, brain-resident innate immune cells, activation accompanied by metabolic reprogramming from OXPHOS to glycolysis (*Baik et al., 2019*). The KEGG pathway hsa00010: Glycolysis/Gluconeogenesis was not enriched in any of our curated DEG sets, but was mildly enriched in IGN-HC; no KEGG pathway related to microglia or inflammation surfaced in our work.

## Novel genes for the KEGG AD pathway

It is curious that AD was less type 1 disrupted than HD and PD in each of HC/PC/SFG (Table S12). Twenty-two of the 54 ALZ-only genes were already in the KEGG AD pathway gene set (Table 2). This set included a single non-mitochondrial gene, *SDHA*; the others were all mitochondrial from OXPHOS. Of the remaining 32 genes, or novel AD genes, 4 were ATP genes from OXPHOS, 14 were from Proteasome, and 14 from *E. coli*, *V. cholerea*, Pyruvate, and TCA; the 28 non-OXPHOS genes include all the non-mitochondrial genes except *SDHA* in the ALZ-only set (Table 2). In comparison to these 28 non-mitochondrial genes, there were only 8 non-mitochondrial AD pathway hits—*SDHA*, *ATF6*, *BACE1*, *CALM1*, *GAPDH*, *GSK3B*, *NAE1*, and *RTN3*—(Table S19) in the ALZ-DEG datasets. In other words, as far as non-mitochondrial genes are concerned the KEGG AD pathway and our ALZ DEG sets are complementary; the former is rich in signal-transduction related genes but lack genes from *E. coli*, *V. cholerea*, Pyruvate, TCA, Proteasome, whereas the latter is *vice versa*. When the 32 novel AD genes were added to the KEGG AD pathway set, then, as expected, AD became more type 1 and type 2 disrupted than HD and PD in all ALZ datasets (Fig. 6, Table S20).

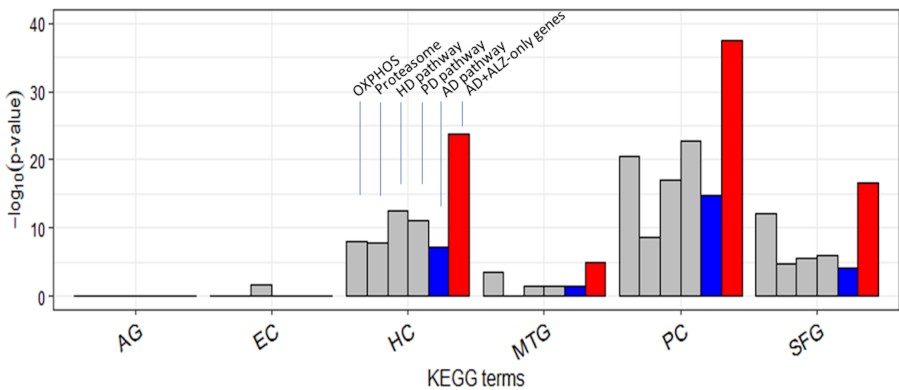

**Figure 6  Enrichment of curated DEG sets in five key KEGG pathways.** In the tower labeled "AD+ALZ-only genes", the gene list of the KEGG AD pathway was augmented with 32 of the 54 ALZ-only genes not already in the list, 14 of which are from the pathway Proteasome. Abbreviation: OXPHOS, hsa00190: Oxidative phosphorylation; Proteasome, hsa03050: Proteasome; PD, hsa05012: Parkinson's disease; HD, hsa05016: Huntington's disease; AD, hsa05010: Alzheimer's disease; AG, normal aging; EC, entorhinal cortex; HC, hippocampus; MTG, medial temporal gyrus; PC, posterior cingulate; SFG, superior frontal gyrus.

In summary, our analysis revealed strong heterogeneity in AD signatures among the five brain regions; HC/PC/SFG showed clear and pronounced AD signatures, MTG moderately so, and EC showed essentially none. There were stark differences between ALZ and AG. Whereas OXPHOS and Proteasome were the most disrupted pathways in HC/PC/SFG, AG showed no OXPHOS disruption and only type 2 Proteasome disruption in AG. Metabolic related pathways including TCA cycle and Pyruvate metabolism were disrupted in ALZ but not in AG. Our result is consistent with the notion that OXPHOS dysfunction is closely related to Aβ and tau pathologies and with the Antimicrobial Protection Hypothesis of AD in that three pathogens infection related pathways were disrupted in ALZ. Many cancer and signaling pathways were disrupted in AG but our results suggest that having AD does not exacerbate such disorders in the aged. We identified 54 "ALZ-only" differentially expressed genes, all down regulated and which, when used to augment the gene list of the KEGG AD pathway, made it significantly more AD-specific. Because OXPHOS dysfunction is directly tied to mitochondrial and oxidative stress disorder and energy supply to brain cells, our result suggests the potential of monitoring the level of neuron activity as a non-invasive method for detecting the possible early onset of AD in the aged, and that maintaining healthy mitochondria and proteasome may be a worthwhile strategy for preventing the onset, or slowing or arresting the progress of AD.

## Limitations and perspectives

The general applicability of our results to all AG and AD cases is limited by the relatively small number of microarray samples used in the present study. Future larger-scaled studies will determine whether our decision to analyze subsets of the original AG and AD datasets with better test-control separations was a wise one. Even though our results are generally consistent with experiments reported in the literature (which we cited), it still lacks direct

experimental support. Hence our results should best be viewed as possible topics for experimental study and verification, chief among which is the clear AG-AD difference with respect to mitochondria and proteasome dysfunction. We may collaborate with experimental researchers in initiating related verification projects. We are interested in joining the Oskar Fischer Project and have AD related discussion with other researchers.

### Abbreviations

| | |
|---|---|
| **AD/ALZ** | Alzheimer's disease; AD for generic use, ALZ when specifically referring to the AD datasets in this study |
| **AG** | aging |
| **IGN** | interacting gene network |
| **xIGN** | xIGN extended IGN |
| **DCE** | differentially co-expressed |
| **DEG** | differentially expressed genes |
| **EC** | entorhinal cortex |
| **GEO** | Gene Expression Omnibus |
| **GOC** | gain of co-expression |
| **HC** | hippocampus |
| **HD** | Huntington's disease |
| **HPRD** | Human Protein Reference |
| **IDCE** | pair interacting DCE pair |
| **IIP** | innate immune pathway |
| **KEGG** | Kyoto Encyclopedia of Genes and Genomes |
| **LOC** | loss of co-expression |
| **MTG** | medial temporal gyrus |
| **PC** | posterior cingulate |
| **PD** | Parkinson's disease |
| **PPI** | protein-protein interaction |
| **SFG** | superior frontal gyrus |
| **Type 1 disruption (of function)** | function represented by a KEGG pathway is enriched in DEGs |
| **Type 2 disruption (of function)** | function represented by a KEGG pathway is enriched in IGN genes |

### Funding

This work was supported by the following grants: MOST-105-2314-B-033-001 from the Ministry of Science and Technology (ROC), CGH-MR-A10506 from Cathay General Hospital, Taiepi, Taiwan, and NCU-LSH-104-A-005, from the National Central University and Landseed Hospital Collaborative, Taoyuan, Taiwan. There was no additional external funding received for this study. The funders had no role in study design, data collection and analysis, decision to publish, or preparation of the manuscript.

## Grant Disclosures

The following grant information was disclosed by the authors:
Ministry of Science and Technology (ROC): MOST-105-2314-B-033-001.
Cathay General Hospital, Taiepi, Taiwan: CGH-MR-A10506.
National Central University and Landseed Hospital Collaborative, Taoyuan, Taiwan: NCU-LSH-104-A-005.

## Competing Interests

The authors declare there are no competing interests.

## Author Contributions

- Yi-Shian Peng conceived and designed the experiments, performed the experiments, analyzed the data, prepared figures and/or tables, authored or reviewed drafts of the paper, and approved the final draft.
- Chia-Wei Tang performed the experiments, analyzed the data, prepared figures and/or tables, and approved the final draft.
- Yi-Yun Peng performed the experiments, prepared figures and/or tables, and approved the final draft.
- Hung Chang analyzed the data, prepared figures and/or tables, and approved the final draft.
- Chien-Lung Chen conceived and designed the experiments, authored or reviewed drafts of the paper, funding acquisition, and approved the final draft.
- Shu-Lin Guo conceived and designed the experiments, authored or reviewed drafts of the paper, funding acquisition, investigation, and approved the final draft.
- Li-Ching Wu performed the experiments, authored or reviewed drafts of the paper, methodology, investigation, formal analysis, supervision, and approved the final draft.
- Min-Chang Huang performed the experiments, authored or reviewed drafts of the paper, funding acquisition, resources, and approved the final draft.
- Hoong-Chien Lee conceived and designed the experiments, authored or reviewed drafts of the paper, overall supervision and funding acquisition, and approved the final draft.

## Data Availability

The raw data is available in the Supplementary Files and at FigShare: Peng, Yi-Shian (2020): PeerJ_Supplementary Materials. figshare. Dataset. https://doi.org/10.6084/m9.figshare.8952938.v1.

## Supplemental Information

Supplemental information for this article can be found online at http://dx.doi.org/10.7717/peerj.8682#supplemental-information.

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
