# Peer review of "Comparative functional genomic analysis of Alzheimer’s affected and naturally aging brains"

_PeerJ, doi:10.7717/peerj.8682_

## Round 0.1 · original submission · Minor Revisions

· Academic Editor

Minor Revisions

Minor revisions as outlined by Reviewer 1 are required.

Reviewer 1 ·

Basic reporting

no comment

Experimental design

no comment

Validity of the findings

no comment

Additional comments

This study describes transcriptomic changes that occur with aging and correlates these changes with AD pathogenesis. Using available microarray databases, Peng et al. mined the AD and Aging microarray results using a comprehensive method to analyze genes involved in AD and aging. Regional differences in differentially expressed gene categories were observed in aging/AD brain. Of interest, the authors observe perturbations in genes related in microbial infection in aging and AD. It is highlighted that the authors performed interacting gene network (IGN) analysis and the extended IGNs to establish a larger view of the transcriptomic landscape in aging/AD.

As written, the manuscript requires some revision. More specifically, the data/figures need to be reorganized and presented to enhance clarity, and the data requires further interpretation and analysis to reinforce the main conclusions in this study.

Major issues:

1. As the authors also mentioned in the “limitation” section, mining results from published microarray data is potentially informative. However, the manuscript presents an abundance of data/figures, but draws few or no conclusions from the analysis. Further clarification, interpretation would enhance the submission.

2. Providing the DEG, IGN, and xIGN in each dataset would provide a good resource to the research community. However, some data refinement may be beneficial: for example, the authors only consider the significance/FDR cutoffs, but fold change was not used to filter the data. Most of the results were generated based on these filtered gene sets, where fold change has not been factored into the results.

3. As described in the methods and Supplementary Figure 2, and Supplementary table 1, the authors selectively retrieved data from the original data sets, which may introduce some bias in the study. It is known that heterogeneity of individuals is a big concern, artificial omission of data in this study requires careful and a detailed explanation.

Minor issues:

1. Line 132 and 136, what is the criteria to selectively analyze the top 106 genes from the AlzGene gene set and top 109 genes in AlzBase gene set?

2. Some of the results were very vague as described. For instance, in line 203-204: “The overall ratio of downregulated to upregulated genes was about 3:1”, which is confusing. The groups compared need to be described.

3. Supplementary Table 1. GSE5281 and GSE53890 should be switched.

4. It is recommended to use several Venn diagrams to show overlap [in categories] rather than overlapping number of genes among DEG, IGN, and xIGN, and between these 6 gene sets (5 AD regions and 1 Aging case).

5. Please carefully check all the citations. Several references were missed, for instance, the source of the aging data set is from GSE53890 (PMID: 24670762) from Lu et al., 2014 have not been cited. The same for GSE5281 (PMID: 17077275, 18332434, 29937276). Since these studies also used the same microarray data, their results need to be compared to the current study. It is nice work to show the connection of OXPHOS and AD, which is one of the highlights in this manuscript. Please discuss more OXPHOS in AD (PMID: 20217279, 31257151), these two articles include the role of OXPHOS in tau pathology and in microglia. Line 400-401, NDUFS3 is also associated with Parkinson’s disease (PMID: 28062948).

Reviewer 2 ·

Basic reporting

The data and background information is outdated.

Experimental design

Too small dataset size and current practices are geared towards reporting the biological processes affected. Individual genes are not that interesting from an in-silico perspective.

Validity of the findings

Nothing new in the findings. Reporting individual genes without wet-lab validation on such outdated datasets that have been heavily worked on is just re-inventing the wheel without anything substantially new to reveal.

Additional comments

Please consider using new datasets with greater sample size or implementing techniques to increase the power of small datasets or new methodologies. The 21 novel genes refer to biological pathways already indicated to be affected in AD. I do not see exactly what is the contribution of this research.

---

## Round 0.2 · accepted · Accept

· Academic Editor

Accept

The revised manuscript is acceptable.